

# Molecular phylogenetics, seed morphometrics, chromosome number evolution and systematics of European *Elatine* L. (Elatinaceae) species

Gábor Sramkó[1,2], Attila Molnár V.[1,2], János Pál Tóth[3], Levente Laczkó[1], Anna Kalinka[4], Orsolya Horváth[1], Lidia Skuza[4], Balázs András Lukács[5] and Agnieszka Popiela[6]

[1] Department of Botany, University of Debrecen, Debrecen, Hungary
[2] MTA-DE "Lendület" Evolutionary Phylogenomics Research Group, Debrecen, Hungary
[3] MTA-DE "Lendület" Behavioural Ecology Research Group, University of Debrecen, Debrecen, Hungary
[4] Molecular Biology and Biotechnology Center, Department of Cell Biology, University of Szczecin, Szczecin, Poland
[5] MTA Centre for Ecological Research, Danube Research Institute, Department of Tisza Research, Debrecen, Hungary
[6] Department of Botany and Nature Conservation, University of Szczecin, Szczecin, Poland

Corresponding author
Attila Molnár V.,
mva@science.unideb.hu

## ABSTRACT

The genus *Elatine* contains *ca* 25 species, all of which are small, herbaceous annuals distributed in ephemeral waters on both hemispheres. However, due to a high degree of morphological variability (as a consequence of their amphibious life-style), the taxonomy of this genus remains controversial. Thus, to fill this gap in knowledge, we present a detailed molecular phylogenetic study of this genus based on nuclear (rITS) and plastid (*accD-psaI*, *psbJ-petA*, *ycf6-psbM-trnD*) sequences using 27 samples from 13 species. On the basis of this phylogenetic analysis, we provide a solid phylogenetic background for the modern taxonomy of the European members of the genus. Traditionally accepted sections of this tree (i.e., *Crypta* and *Elatinella*) were found to be monophyletic; only *E. borchoni*—found to be a basal member of the genus—has to be excluded from the latter lineage to achieve monophyly. A number of taxonomic conclusions can also be drawn: *E. hexandra*, a high-ploid species, is most likely a stabilised hybrid between the main sections; *E. campylosperma* merits full species status based on both molecular and morphological evidence; *E. gussonei* is a more widespread and genetically diverse species with two main lineages; and the presence of the Asian *E. ambigua* in the European flora is questionable. The main lineages recovered in this analysis are also supported by a number of synapomorphic morphological characters as well as uniform chromosome counts. Based on all the evidence presented here, two new subsections within *Elatinella* are described: subsection *Hydropipera* consisting of the temperate species of the section, and subsection *Macropodae* including the Mediterranean species of the section.

## INTRODUCTION

Waterworts (genus *Elatine* L.; Elatinaceae, Malpighiales) are small, ephemeral, aquatic herbaceous annuals (Fig. 1), or short-lived perennials, inhabiting the muddy surfaces of ephemeral waters (e.g., temporary pools, shores of lakes and ponds, marshes, and rice-fields). These plants have an interrupted but cosmopolitan distribution, showing strong preference for temperate regions in middle and high latitudes as well tropical mountain ranges (e.g., the Andes). The small, inconspicuous, and mostly cleistogamous flowers of waterworts are usually self-pollinating, but outcrossing can also take place (i.e., facultative autogamy). Since no recent monograph exists for this genus, the total number of species is thought to be between *ca* 10 (*Kubitzki, 2014*) to *ca* 25 (*Tucker, 1986*). Most of the species is found in Europe, where ten species is registered (*Uotila, 2009b*) although Flora Europaea lists only eight species (*Cook, 1968*). Another center of the genus is in North America, where nine species are present (*Tucker, 1986*).

Surprisingly little work has been completed recently in the taxonomy of *Elatine*. The most recent worldwide monograph (*Niedenzu, 1925*) echoes the earlier work of *Seubert (1845)*, and although *Moesz (1908)* proposed a slightly different classification, the original iteration is still used for the taxonomy of this genus. According to this classification, the genus can be split into two subgenera, *Potamopytis* (Adanson) Seub. which is represented just by the morphologically distinct (leaves in whorls) species *Elatine alsinastrum* L., and subgenus *Elatine* Seub. (subg. *Hydropiper* Moesz) which contains all the other species (leaves arranged opposite). The *Elatine* subgenus is further divided into two sections: *Elatinella* Seub., which includes species with diplostemonous flowers (i.e., stamens arranged in two whorls and thus having double the number of sepals), usually arranged in a tetramerous flower; and *Crypta* (Nutt.) Seub., which includes species of trimerous flowers that show haplostemony (i.e., an arrangement of stamens in a single whorl opposite the sepals thus having an equal number of anthers and sepals). While Europe is rich in species belonging to section *Elatinella* (all species included in this section are native to Europe and temperate Asia with the exception of the North American *E. californica* A. Gray and South American *E. ecuadoriensis* Molau), section *Crypta* has a center of species diversity in North America, while Eurasia boasts just two species, *E. ambigua* Wight and *E. triandra* Schkuhr. The species of waterworts that occur in the Southern Hemisphere are all members of the latter section, with the exception of *E. ecuadoriensis*. The work presented in this paper focuses on the European species but also provides an outlook on the North American members of the genus.

Although recent work has augmented our knowledge of the biology of European *Elatine* (*Popiela & Łysko, 2010*; *Popiela et al., 2011*; *Popiela et al., 2012*; *Molnár, Popiela & Lukács, 2013b*; *Popiela et al., 2013*; *Takács et al., 2013*; *Kalinka et al., 2014*), there are still few studies that deal with this taxonomy of this genus in Europe (*Mifsud, 2006*; *Uotila, 2009c*; *Molnár et al., 2013a*). In the meantime, many new species have been described from the Americas and Australia (*Mason, 1956*; *Schmidt-Mumm & Bernal, 1995*; *Albrecht, 2002*; *Garneau, 2006*; *Lægaard, 2008*). Most researchers agree that seed morphology is of exceptional importance in the taxonomy of *Elatine* (*Moesz, 1908*; *Mason, 1956*; *Cook, 1968*; *Mifsud,*

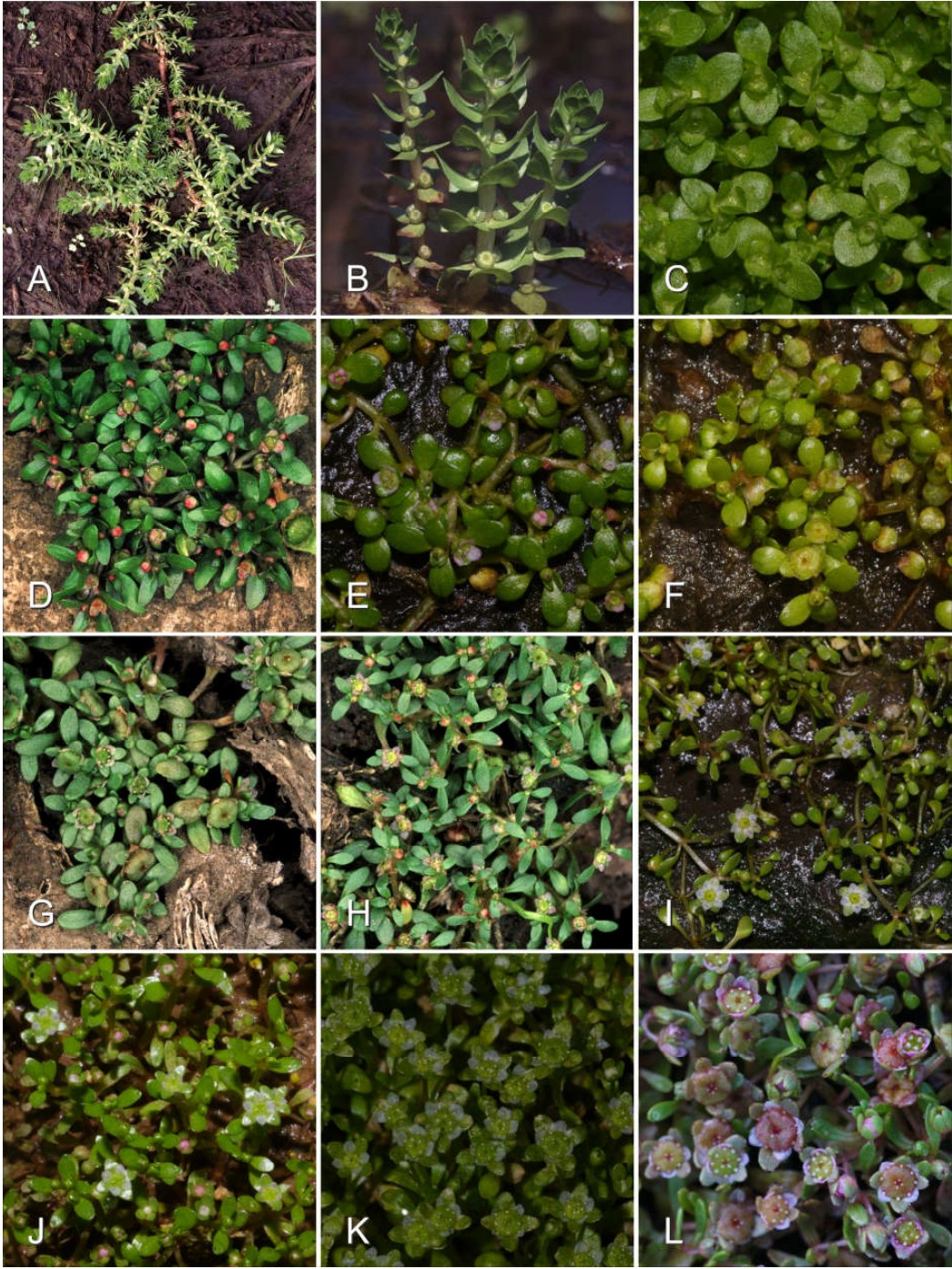

**Figure 1  Examples of morphological diversity in the genus _Elatine_.** (A) Habit of the terrestrial form of _Elatine alsinastrum_ (_E._ subgenus _Potamopithys_); (B) flowering shoots of water-living form of _Elatine alsinastrum_; (C) _E. brochonii_ (Spain); (D) _E. triandra_ (Hungary); (E) _E. hexandra_ (Poland); (F) _E. californica_ (USA); (G) _E. hungarica_ (Hungary); (H) _E. hydropiper_  (Hungary); (I) _E. campylosperma_ (Spain); (J) _E. gussonei_ (Lampedusa); (K) _E. gussonei_ (Sicily); (K) _E. macropoda_ (Sardinia). Photographs by A. Molnár V.

*2006*; *Molnár et al., 2013a*; *Molnár et al., 2015*), and seed shape (i.e., how much it is curved) as well as seed surface reticulation (i.e., number and shape of seed pits) have traditionally been characters of high significance used for recognising species in the genus (*Moesz, 1908*; *Tucker, 1986*; *Mifsud, 2006*; *Molnár et al., 2013a*). Although vegetative characters, including pedicel length and leaf-shape, are also sometimes emphasised as important sources of taxonomic information (*Seubert, 1845*; *Niedenzu, 1925*), these features have generally been thought to be more variable between aquatic and terrestrial forms of the same species than between separate species (*Mason, 1956*; *Molnár et al., 2013a*; *Molnár, Popiela & Lukács, 2013b*; *Molnár et al., 2015*).

Over recent decades, this genus has received a great deal of attention from molecular phylogenetic workers following the discovery of its important phylogenetic position within Malpighiales (*Davis & Chase, 2004*). Indeed, the bulk of studies dealing with this order have paid much attention to samples of *Elatine* as representative of the family (*Davis et al., 2005*; *Tokuoka & Tobe, 2006*), and only a very recent one focused on internal phylogenetic relationships within the genus (*Cai et al., 2016*). Most recent studies have clearly indicated a sister-group relationship between the two genera (i.e., *Elatine* and *Bergia* L.) in family Elatinaceae (*Korotkova et al., 2009*; *Wurdack & Davis, 2009*; *Davis & Anderson, 2010*; *Xi et al., 2012*), and a divergence age of 85–113 million year ago (Ma) has estimated for this lineage (*Davis et al., 2005*). In spite of this phylogenetic information, at the time of writing this paper we knew almost nothing about phylogenetic relationships within the genus *Elatine*, and so were unable to test the taxonomic hypothesis of *Seubert (1845)* postulated more than a century and a half ago.

To fill this gap in our knowledge, we present a molecular phylogeny of the genus *Elatine* in this paper that employs 27 samples from 13 species based on nuclear and plastid sequences. In addition to molecular work, we also present a morphometric analysis of seeds that enables a test of the explanatory power of seed morphology as a phylogenetic took within this genus. Our aim is to test taxonomic treatments that are currently applied as well as to provide a modern systematic treatment based on our results.

## MATERIALS AND METHODS

### Plant material and taxon sampling

Plant tissue samples were collected and embedded in silica-gel from cultivated plants kept in Debrecen that encompass the taxonomic range of genus *Elatine* in Europe (Table 1). *Elatine alsinastrum, Elatine hungarica*, *E. hydropiper* and *E. triandra* are protected species and were sampled in Hungary with the permission of the Hortobágy National Park Directorate (Permission id.: 45-2/2000, 250-2/2001). Although our focus was on this region, represented by all 11 species found on the continent (i.e., four from section *Crypta*, six from section *Elatinella*), we included samples from North American members of the genus, and added *Bergia texana* as an outgroup. Altogether, 14 species are represented in our collection covering almost the whole section *Elatinella* (only *E. ecuadoriensis* is missing), and four species of section *Crypta* are also included. Cultivated plants originate from our field collections of seedling plants or seeds, sown on sterile soil and kept in climate-controlled

chambers. Thus, 1–5 young plants from germinated seeds were grown under constant conditions (14 h/day light and 30 μmol m-2 sec-1 light intensity; temperatures: daytime 22 ± 2 °C, dark hours 18 ± 2 °C) to form clonal groups and in order to set seed. Plant material of a single discrete clonal group was sampled into silica-gel for DNA analysis, while mature seeds were collected from plants. These seeds were sent to the Polish co-authors in Szczecin for scanning electron microscopy (SEM).

## DNA regions considered

Since at the time of designing our study no previous molecular work has ever discovered the intra-generic molecular variability of the genus *Elatine*, we screened one mitochondrial region, three nuclear regions, and nine plastid regions commonly used in plant phylogenetics on three selected samples (Table S1). Of these, the nuclear ribosomal ITS (nrITS) region (*Baldwin et al., 1995*; *Álvarez & Wendel, 2003*; *Nieto-Feliner & Rosselló, 2007*), the *accD-psaI* intergenic spacer (*Small et al., 1998*), the *psbJ-petA* intergenic spacer (*Shaw et al., 2007*), and the *ycf6-psbM-trnD* intergenic spacer (*Shaw et al., 2005*), the latter three regions representing the plastid genome, were chosen for sequencing across the whole sample set (see 'Results').

## DNA extraction, amplification, cloning, and sequencing

Total genomic DNA was extracted from approximately 15–30 mg of silica-gel dried plant material, thoroughly ground in liquid nitrogen and then resuspended in lysis buffer (2% CTAB, 20 mM EDTA pH 8, 100 mM Tris–HCl pH 9, and 1.4 mM NaCl). Following incubation at 65 °C for 60 min, samples were centrifuged at 20,000 g for 3 min, before supernatant was extracted with an equal volume of chloroform and centrifuged for 10 min at 20,000 g. This extraction procedure was repeated twice, and DNA was precipitated with an equal volume of iso-propanol plus 0.08 volume of 7.5 M ammonium-acetate and stored at −20 °C for 1 h. DNA was pelleted by centrifugation at 20,000 g for 12 min; each pellet was washed twice with 70% ethanol, dried on open-air, and redissolved in 40–100 μl 0.1 M Tris (pH 7.5).

The angiosperm-specific ITS1A (5′- GAC GTC GCG AGA AGT CCA) primer (*Gulyás et al., 2005*) and the universal primer ITS4 (*White et al., 1990*) were applied for polymerase chain reaction (PCR) to specifically amplify plant nrITS. The PCR reaction mixture contained 0.1 volume 10 × Taq buffer with $(NH_4)_2SO_4$ (Fermentas), 200 μM of each dNTPs (Fermentas), 2 mM $MgCl_2$, 0.2 μM of each primer, 1.25 U DreamTaq Green polymerase (Fermentas), and 1 μl unquantified genomic DNA extract. Amplifications were performed on an Abi Veriti 9600 thermal cycler (Applied Biosystems), programmed for an initial denaturation step at 94 °C for 4.30 min, followed by 33 cycles of denaturation for 30 s at 94 °C, annealing for 30 s at 51 °C, and extension for 30 s at 72 °C. Extension times were increased by one second at each cycle, and thermal cycling ended with a final extension for 7.00 min at 72 °C.

All plastid regions were amplified by the primers described in their corresponding publications and under the same PCR conditions. The reaction mixture was the same as described for nrITS, and the amplification regime followed a touchdown protocol: an

Srankó et al. (2016), *PeerJ*, DOI 10.7717/peerj.2800

**Table 1** The samples included in this study; species sampled, sample origins, abbreviated sample names (as appear on phylogenetic trees) and GenBank accession numbers.

| Species | Locality | Lat. (°N) | Long. (°E) | Abbreviated name | GenBank accession numbers | | | |
|---|---|---|---|---|---|---|---|---|
| | | | | | nrITS | accD-psaI | psbJ-petA | ycf6-psbM-trnD |
| Elatine alsinastrum L. | Hungary: Tiszalúc | 48.03 | 21.11 | E. alsinastrum (HU) | KX555572 | KX818143 | KX818170 | KX818116 |
| E. ambigua Wight | Italy: Vigevano | 45.33 | 8.79 | E. ambigua (IT) | KX555573 | KX818150 | KX818177 | KX818123 |
| E. ambigua Wight | Nepal: Aardash Nagar | 27.01 | 84.86 | E. ambigua (NP) | KX555574 | KX818151 | KX818178 | KX818124 |
| E. brachysperma A.Gray | USA, California: Fallbrook | 33.46 | −117.37 | E. brachysperma (US) | KX555575 | KX818146 | KX818173 | KX818119 |
| E. brochonii Clav. | Morocco: Ben Slimane | 33.62 | −7.07 | E. brochonii (MA) | KX555576 | KX818144 | KX818171 | KX818117 |
| E. brochonii Clav. | Spain: San Silvestre de Guzmán | 37.4 | −7.36 | E. brochonii (SP) | KX555577 | KX818145 | KX818172 | KX818118 |
| E. californica A.Gray | USA, California: Los Angeles | 33.82 | −118.34 | E. californica (US) | KX555578 | KX818154 | KX818181 | KX818127 |
| E. campylosperma Seub. ex Walp. | Italy: Sardinia: Gesturi | 39.73 | 9.03 | E. campylosperma (IT) | KX555579 | KX818160 | KX818187 | KX818133 |
| E. campylosperma Seub. ex Walp. | Spain: El Rocío | 37.12 | −6.49 | E. campylosperma (SP) | KX555580 | KX818161 | KX818188 | KX818134 |
| E. gussonei (Sommier) Brullo et al. | Morocco: Ben Slimane | 33.61 | −7.1 | E. gussonei (MA) | KX555581 | KX818163 | KX818190 | KX818136 |
| E. gussonei (Sommier) Brullo et al. | Spain: Casar de Cáceres | 39.33 | −6.25 | E. gussonei (SP) | KX555582 | KX818168 | KX818195 | KX818141 |
| E. gussonei (Sommier) Brullo et al. | Malta: Gózó: Ta' Sannat | 36.01 | 14.25 | E. gussonei (MT) | KX555583 | KX818164 | KX818191 | KX818137 |
| E. gussonei (Sommier) Brullo et al. | Italy: Lampedusa | 35.51 | 12.56 | E. gussonei (LMP) | KX555584 | KX818169 | KX818196 | KX818142 |
| E. gussonei (Sommier) Brullo et al. | Italy: Sicily: Modica | 36.76 | 14.77 | E. gussonei (IT) | KX555585 | KX818162 | KX818189 | KX818135 |
| E. hexandra DC. | Spain: San Silvestre de Guzmán | 37.4 | −7.36 | E. hexandra (SP) | KX555586 | KX818148 | KX818175 | KX818121 |
| E. hexandra DC. | Poland: Parowa | 51.39 | 15.23 | E. hexandra (PL1) | KX555587 | Not included | Not included | Not included |
| E. hexandra DC. | Poland: Poznań (Milicz) | 51.55 | 17.35 | E. hexandra (PL2) | KX555588 | KX818147 | KX818174 | KX818120 |

**Table 1** (*continued*)

| Species | Locality | Lat. (°N) | Long. (°E) | Abbreviated name | GenBank accession numbers | | | |
|---------|----------|-----------|------------|------------------|---------|---------|---------|---------|
| | | | | | nrITS | *accD-psaI* | *psbJ-petA* | *ycf6-psbM-trnD* |
| *E. hexandra* DC. | UK: Cornwall, Bodmin Moor[a] | NA | NA | *E. hexandra* (GB) | KX555589 | KX818149 | KX818176 | KX818122 |
| *E. hungarica* Moesz | Hungary: Konyár | 47.31 | 21.67 | *E. hungarica* (HU) | KX555590 | KX818155 | KX818182 | KX818128 |
| *E. hungarica* Moesz | Russia: Volgograd | 49.76 | 45.7 | *E. hungarica* (RU) | KX555591 | KX818156 | KX818183 | KX818129 |
| *E. hydropiper* L. | Hungary: Tiszagyenda | 47.36 | 20.52 | *E. hydropiper* (HU) | KX555592 | KX818157 | KX818184 | KX818130 |
| *E. hydropiper* L. | Poland: Kwiecko | 54.03 | 16.69 | *E. hydropiper* (PL) | KX555593 | KX818158 | KX818185 | KX818131 |
| *E. macropoda* Guss. | Turkey: Büyükhusun | 39.51 | 26.38 | *E. macropoda* (TR) | KX555594 | KX818166 | KX818193 | KX818139 |
| *E. macropoda* Guss. | Spain: Casar de Cáceres | 39.19 | −6.29 | *E. macropoda* (SP) | KX555595 | KX818165 | KX818192 | KX818138 |
| *E. macropoda* Guss. | Italy: Sardegna: Olmedo | 40.63 | 8.41 | *E. macropoda* (IT) | KX555596 | KX818167 | KX818194 | KX818140 |
| *E. orthosperma* Dueb. | Finland: Oulu | 65.06 | 25.47 | *E. orthosperma* (FI) | KX555597 | KX818159 | KX818186 | KX818132 |
| *E. triandra* Schkuhr | Poland: Janików | 51.56 | 14.98 | *E. triandra* (PL) | KX555598 | KX818153 | KX818180 | KX818126 |
| *E. triandra* Schkuhr | Hungary: Karcag | 47.27 | 20.9 | *E. triandra* (HU) | KX555599 | KX818152 | KX818179 | KX818125 |
| *Bergia texana* Seub. ex Walp.[b] | USA, California: Perris | 33.84 | −117.13 | Bergia | KX555600 | Not included | Not included | Not included |

**Notes.**
[a]purchased from Kew DNA Bank (ID: 12361).
[b]included as outgroup.

initial denaturation step at 94 °C for 5 min, followed by 20 cycles of denaturation for 30 s at 94 °C, annealing for 30 s starting at 58 °C, decreased by 0.5 °C in each cycle, then extension for 1.30 s at 72 °C, followed by 20 subsequent cycles using the same regime but keeping the annealing temperature constant at 48 °C. Thermal cycling was ended with a final extention at 72 °C for 7.00 min.

Quality and quantity of PCR products were evaluated by loading them onto a 1% agarose gel stained with ethidium bromide. For direct sequencing, unpurified PCR products were submitted to a commercial purification and sequencing service provided by Macrogen Inc. (Korea). All regions were sequenced from the forward and reverse directions using the original primers as sequencing primers. The *ycf6-psbM-trnD* region was amplified as a whole, but sequenced using the *psbM* gene anchoring primers as additional internal sequencing primers.

Cloning was performed on two *E. hexandra* samples from Poland which showed unambiguous double-peaks in their nrITS direct sequences. These purified PCR products were ligated and transformed into the pGEM-T Easy Vector System II (Promega) following the manufacturer's instructions. Transformed cells were screened with ampicillin, and recombinant plasmid DNA was isolated from white colonies by suspending them in 40 μl sterile MilliQ water, subboiling for 5 mins at 98 °C, and then centrifugating at 20,000 g. The plant nrITS region was amplified and cycle-sequenced from eight and ten clones per individual in the same PCR and sequencing conditions as described above. Clone sequences were sequenced only from the forward direction.

## Sequence analyses, alignment and phylogenetic analyses

Forward and reverse sequencing reads were manually checked by eye using the software Chromas Lite v.2.01 (Technelysium Pty). The nrITS sequences were carefully screened for additive polymorphic sites (i.e., overlapping peaks at certain, phylogenetically informative sites), and IUPAC ambiguity symbols were used to indicate these when two nucleotides occurred together at the electropherogram rather than an indication of ambiguous reading. In one case (*E. gussonei* MT)—where the additivity of a single site in the nrITS sequence was evidently from the result of two, closely related species—we used the sequences of the supposed parental species (*E. gussonei* LMP and *E. macropoda* IT) to represent this sample in our analyses. Sequences were aligned manually using BioEdit v.7.1.3 (*Hall, 1999*). Due to the great number of mutations separating *Bergia texana* from *Elatine*, we were unable to reliably align it in the plastid matrix. Consequently, we had to exclude it from the plastid and combined analyses, but we were able to include it in the nrITS matrix. The phylogenetic relationship between the cloned ribotypes of Polish *E. hexandra* samples was inferred with the software TCS v.1.21 (*Clement, Posada & Crandall, 2000*) using default parameters but allowing the connection of ribotypes 100 steps away. One of the most frequently occurring cloned ribotypes (PL1.3, PL1.10, PL2.1, PL2.7) of each ribogroup was selected to represent (see *LaJeunesse & Pinzón, 2007*) the cloned samples in the phylogenetic analyses. All sequences are deposited in GenBank (Table 1.)

We worked with two main data matrices. One consisted of the nuclear marker (nrITS), the second one of the plastid markers (comprising of *accD-psaI*, *psbJ-petA*, *ycf6-psbM-trnD*).

The latter regions were combined together because the plastid genome is inherited as a unit and is not subject to significant recombination (*Palmer et al., 1988*; *Jansen & Ruhlman, 2012*), making it readily combinable for phylogenetic analyses (*Doyle, 1992*; *Soltis & Soltis, 1998*). Given the high number of variable nucleotide sites in the above two matrices, gaps were treated as missing in subsequent analyses. Following separated analyses of the two main matrices, we checked for 'hard incongruencies' (*Mason-Gamer & Kellogg, 1996*; *Seelanan, Schnabel & Wendel, 1997*; *Wendel & Doyle, 1998*) in the resulting trees: branch placement was only considered to be in hard incongruence when they received >70% bootstrap and >0.95 posterior probability support—an approach advocated by many workers (*Daru et al., 2013*; *Patchell, Roalson & Hall, 2014*; *Scheunert & Heubl, 2014*), favoured over the commonly used ILD-test (*Farris et al., 1994*), which can fail to correctly test combinability (*Dolphin et al., 2000*; *Barker & Lutzoni, 2002*; *Darlu & Lecointre, 2002*). Since no such hard incongruence was observed, we combined the nuclear and plastid matrices into a combined matrix. Samples with nrITS sequences showing paralogy [*hexandra* (PL), *gussonei* (MT)] were only represented by their plastid sequences in this latter matrix (i.e., only their plastid sequences were used in the combined analyses, nrITS was coded as 'missing').

Heuristic searches using the Maximum Parsimony (MP) criterion were conducted on the three matrices (i.e., nuclear, plastid, combined) separately in Paup v.4.0b*10 (*Swofford, 2003*) using the same settings. In addition to default settings, the search utilised a tree bisection-reconnection swapping algorithm holding ten trees in each iteration step with 1,000 random sequence replicates. All most parsimonious trees (MPTs) were saved and an arbitrary chosen tree was interpreted via statistical branch support. The statistical robustness of tree topology was tested via the non-parametric bootstrap procedure (*Felsenstein, 1985*) using 1,000 pseudo-replicates in simple heuristic search. Branches were considered to be none (<50%), weakly (51–74%), moderately (75–84%), or strongly (>85%) supported.

Phylogenetic trees using Bayesian inference (BI) were also constructed using the same matrices. These trees were built using the MrBayes v.3.2.2 software (*Ronquist et al., 2012*) using the model-jumping feature. Thus, various possible models of molecular evolution were sampled for each gene (both single and combined data) during the analysis by taking advantage of command lset applyto = (all) nucmodel = 4by4 nst = mixed rates = gamma covarion = no;'. The combined matrix was partitioned into nuclear and plastid datasets, and these were treated separately during the runs. Two independent Markov chain Monte Carlo analyses with four simultaneous chains (one cold and three heated) for each analysis were run for 10,000,000 generations by sampling trees and parameters in every 1,000th generation, while convergence of the two runs was checked using Tracer v.1.5 (available from http://tree.bio.ed.ac.uk/software/tracer/) inspecting effective sample sizes and visually evaluating the joint-marginal densities and log likelihood traces. We discarded the first 2,500,000 generations as 'burn in' and trees were summarised using the 50 percent majority rule method. Posterior probability (PP) values of each branch were considered as test of statistical robustness treating branches with PP <0.85 as none, 0.85–0.89 as weakly, 0.9–0.95 as moderately, >0.95 as strongly supported.
## Morphometric data collection and analyses

In addition to the molecular phylogenetic data, morphological data were collected on all species analysed here. As has been demonstrated previously (*Moesz, 1908*; *Mifsud, 2006*; *Uotila, 2009c*; *Molnár et al., 2013a*; *Molnár, Popiela & Lukács, 2013b*), the seed morphology of *Elatine* spp. is the most reliable character for species-level taxonomy. To obtain material for the morphometric data acquisition, field-collected seeds of the species were transferred to the laboratory, and mature plants were raised in climatic chambers. Fifty seeds were collected from mother plants, and Scanning Electron Microscopic (SEM) pictures were taken. Fifty seeds from each population were photographed and their outlines were digitised using tpsDig2 (*Rohlf, 2010*). The outline coordinates were transformed using Hangle-Fourier (*Haines & Crampton, 2000*) function using the PAST v.1.7c (*Hammer, Harper & Ryan, 2001*) program, and statistical analyses were carried out based on on Hangle coefficients. In order to determine the relationships between predefined groups, a linear discriminant analysis was conducted such that *a priori* groups were populations. The group centroids were visualised on scatter plots, and Wilks's λ was used to measure the discriminatory power of the model with values changing from 0 (perfect discrimination) to 1 (no discrimination). Classification was made using the Jackknifed grouping function in PAST; this method, one known specimen is left out each iteration, and assigned using the discriminant function which is calculated based on all the cases except that given. The numbers of correct assignments were used to evaluate the usefulness of the discriminant function. High numbers of correct assignments indicate diagnostic differences between the surveyed groups. We also used pair-wise MANOVA to test the statistical significance between groups.

## RESULTS

### Initial screen of sequence variability

The twelve regions initially checked for phylogenetic variability showed (Table S1) the typical situation for plants (*Soltis & Soltis, 1998*); nuclear regions were most variable, followed by plastid intergeneric spacers, and plastid genes, while the mitochondrion-encoded *nad6* gene was the least variable between the outgroup and the two ingroups. Amongst nuclear genes, nrITS was found to be the most variable, while *ycf6-psbM*, *trnL-trnF*, *accD-psaI*, and *psbJ-petA* changed the most of the plastid regions. It is noticeable that *At103* and *Eif3E* showed extensive paralogy in some pilot sequencings of additional *Elatine* material; thus, we excluded these from further work. Although it was amongst the highly variable regions, the *trnH-psbA* sequence was also excluded as this marker produced unreadable direct sequences; some pilot sequences showed abrupt unreadability in the middle portion of the intergeneric spacer, but what was recovered at the other end of the region was as if it had length-different paralogous copies. Finally, the region *psbM-trnD*, albeit its relative invariability, was used, and sequenced, as it was convenient to include together with the highly variable *ycf6-psbM* region.

## Molecular phylogenetic relationships

The parsimony analysis of the nrITS matrix found two equally MPTs of 235 steps with negligible homoplasy (consistency index (CI) = 0.8851, homoplasy index (HI) = 0.1149, retention index (RI) = 0.9403). These trees differ from each other in the main clustering 'above *E. alsinastrum*,' which is not supported by bootstrap analysis and not discussed further. One of the MPTs with support information is displayed as a phylogram (Fig. 2A), with *Bergia texana*, representative of the sister genus, placed at the root. The following branches towards the tips are not resolved; although placed sister to the rest of the genus in the parsimony analysis, the sister relationship of *Elatine alsinastrum* is not supported in subsequent bootstrap analysis. There are four more clades identifiable with statistical confidence; that including *E. brochonii*, *E. hexandra*, *E. brachysperma*, *E. triandra* (including *E. ambigua* and two clones of Polish *E. hexandra*), and the clade of tetramerous species. Within the latter, the temperate representatives of this group are sister to the Mediterranean species, which also split into two clades, one containing most *E. macropoda* and *E. campylosperma* samples, and a second containing all *E. gussonei* samples plus a Turkish *E. macropoda* accession. Within the temperate species of this lineage there is almost no resolution with the exception of the two *E. hydropiper* accessions which are placed as sisters to each other. The tree obtained by BI totally supports this topology as all branches, found to be supported by the MP bootstrap procedure, were also supported by PP values and thus did not collapse on the majority rule consensus tree of BI (not shown). Therefore, only the PP values are shown on the corresponding branches.

The MP analysis of the plastid dataset found two equally MPTs 315 steps in length with negligible homoplasy (CI = 0.9016, HI = 0.0984, RI = 0.9556). One of these, with support information, is displayed as a phylogram (Fig. 2B), but as we were unable to reliably align *Bergia* in this dataset, it was left out from here, and our use of *E. alsinastrum* as an alternative outgroup was not supported. From this split towards the tips the branches usually receive high statistical support, with *E. brochonii* branching off first. This was followed by *E. brachysperma* and *E. triandra* clade. The tetramerous clade is again well-supported, but on the plastid tree (Fig. 2B) we find *E. campylosperma* to be sister to the rest of the group, although this relationship is weakly supported. There is less resolution at the next level of branches, where a strongly supported branch separates *E. macropoda* and two *E. gussonei* (MT and LMP) samples from the rest, then all other *E. gussonei* samples form a clade, *E. hexandra* samples form a clade, finally the temperate members of the tetramerous species form a clade. Unlike on the nrITS tree, there is resolution in the latter clade as the North American *E. californica* branches first, this is followed by *E. orthosperma* what is sister to the *E. hungarica* plus *E. hydropiper* clade, where we find no resolution. Again, the majority rule consensus tree found by BI had a fully corroborative topology (not shown), therefore, only the PP values are shown on the MP phylogram at the corresponding branches.

Regarding testing the species delimitation of *a priori* taxa, multiple accessions of the same species, in most cases, formed strongly supported monophyletic groups. Nevertheless, there were some significant exceptions on the nrITS tree (Fig. 2A), including the Turkish *E. macropoda* sample was clustered together with *E. gussonei* samples, the Spanish

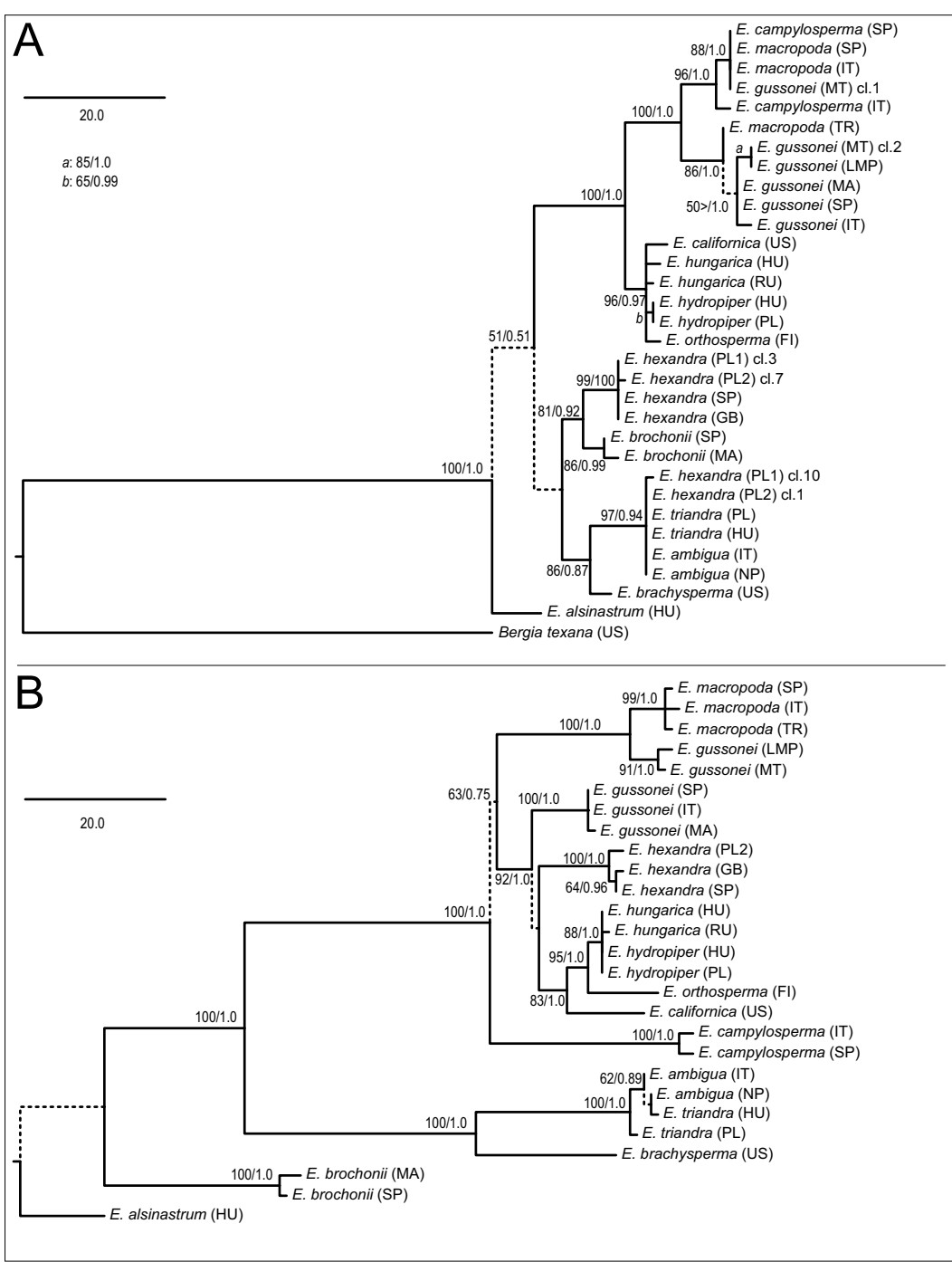

**Figure 2  Phylogenetic trees reconstructed using the nrITS (A) and plastid (B) matrices.** Both trees are arbitrary chosen MPTs displayed as phylograms with bootstrap support percentages/posterior probability values at the corresponding branches. Dash indicates lack of statistical support, and such branches are indicated by dashed lines. A scale bar representing 20 mutational changes is displayed on both trees, and the abbreviation 'cl.' denotes cloned nrITS sequences on the nrITS tree.

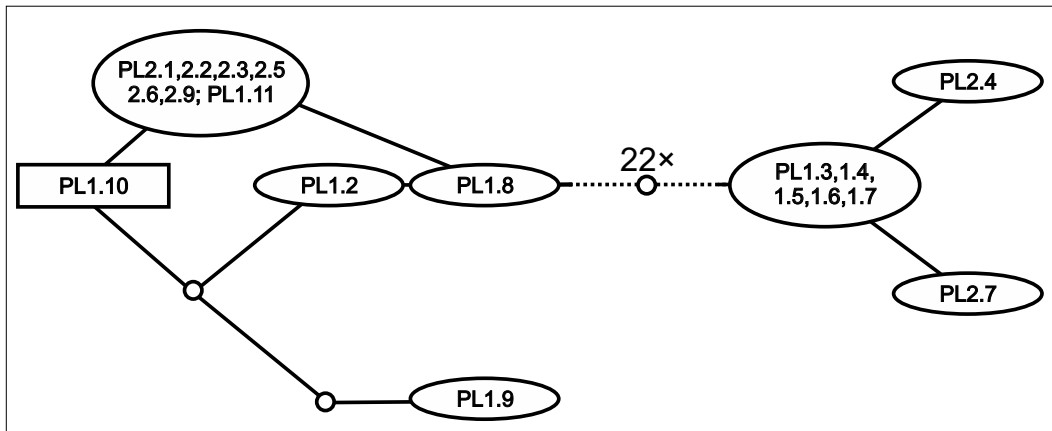

**Figure 3  TCS-network of cloned nrITS sequences of Polish *E. hexandra* accessions.** Clone names follow the abbreviated sample name, while the number following it is the indetifier of the clone sequenced. Hypothetical (unrecovered) ribotypes are represented by small circles, and the 22 such ribotypes separating the two ribotype-groups are not represented to keep the figure easily readable.

*E. campylosperma* sample was clustered together with the *E. macropoda* samples. On the plastid tree only the *E. gussonei* samples fell into two separate clades, with two accessions from Malta and Lampedusa sister to *E. macropoda*, and three accessions of *E. gussonei* from Italy, Spain and Morocco clustered separately.

There were some incongruent placement of branches, but these only affected tip, or close to tip, branches. The most important of these concerns *E. campylosperma*, nested within the Mediterranean clade on the nrITS tree, but placed as sister to the rest of the tetramerous clade on the plastid tree. Secondly, samples of *E. hexandra* are either placed as sister to *E. brochonii* (although with weak support) or together with *E. triandra* on the nrITS tree, while these samples are found nested within the clade comprising the three *E. gussonei* listed above as well as the temperate members of the tetramerous plants.

The TCS-analysis of nrITS clones of Polish *E. hexandra* unravelled the existence of two ribotype-groups within the same individuals (Fig. 3); these two groups were separated by 22 mutation steps. When included in a wide phylogenetic context (Fig. 2A) the representative ribotypes fall into two very distant clades on the nrITS tree; one group of clones are inseparable from *E. triandra* and *E. ambigua* direct sequences, while the other group forms a separate clade together with direct sequences of *E. hexandra*, sister to *E. brochonii*.

When we combined the nrITS and plastid dataset the MP search found a single MPT of 443 steps containing little homoplasy (CI = 0.8375, HI = 0.1625, RI = 0.9322) (Fig. 4). Bootstrap analysis recovered support for all branches but the placement of *E. alsinastrum* at the root, the placement of Polish *E. triandra* as sister to the *E. ambigua* sample, and the relationship between *E. hungarica* samples remained unresolved. In addition, most nodes are highly supported by bootstrap values, and the analysis using BI also found a most credible phylogenetic tree with the very same topology as the MP tree. Again, this tree is not shown, just the PP values are indicated on its counterpart. Multiple accessions of the same *a priori* species form strongly supported clades in the combined dataset, with

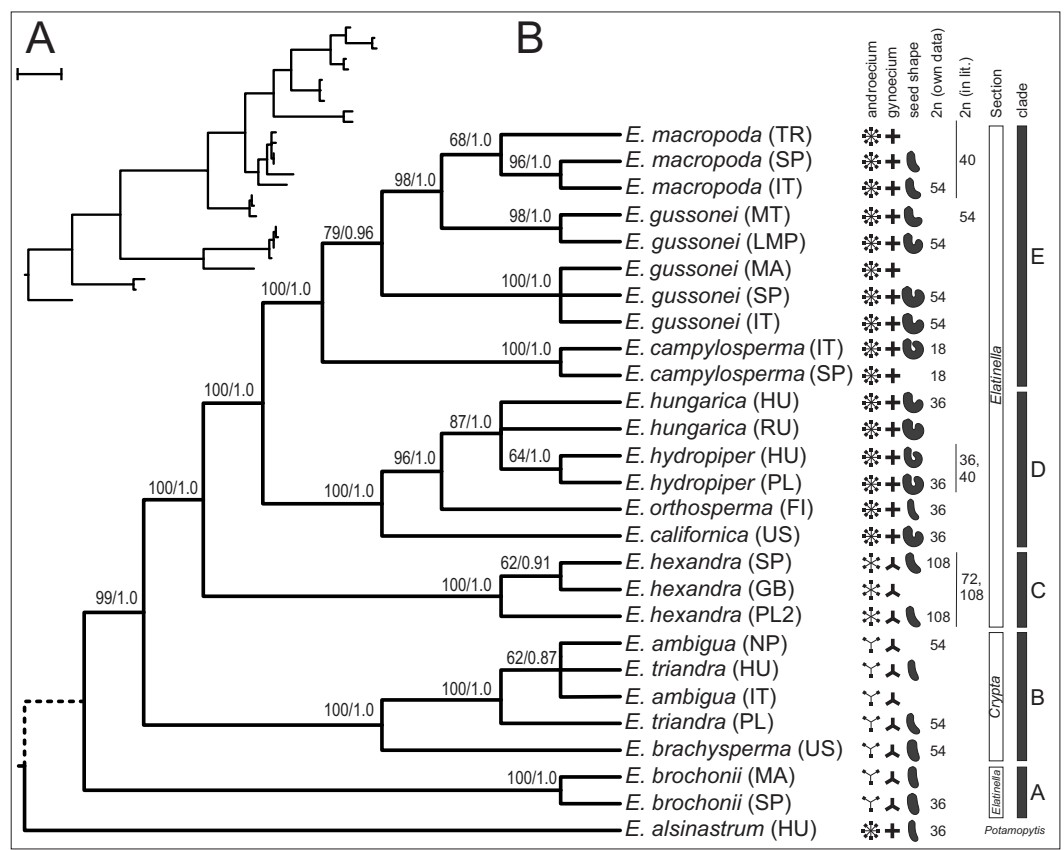

**Figure 4** One of the two MPTs resulted from MP analysis of the combined (nrITS+plastid *accD-psaI*, *psbJ-petA*, *ycf6-psbM-trnD*) sequences displayed as a phylogram (A) and as a cladogram (B). Next to each branch are bootstrap support values resulting from 1,000 pseudo-replicate followed by Bayesian PP values after the slash.

the exception is *E. hungarica*, where samples are not resolved as monophyletic on either tree; another one concerns the samples of *E. gussonei*, which fall into two separate, closely related clades on the trees.

## Morphometric comparison of seeds

Measured populations of different *Elatine* species proved to be significantly different based on seed outlines, a result with high discriminatory power (Wilks's $\lambda = 0.00004$, $p < 0.001$). The first axis explained 75% and the second 12% of variance between groups; on the scatter plot of group centroids four morpho-groups can be recognized, straight (I.), curved (II.), highly curved (III.), and an intermediate (IV.) form between curved and straight (Fig. 5). Surprisingly, the results of the post-hoc test indicated that nearly all predefined groups were significantly different from each other except the two *Elatine hexandra* populations (Table S2). The cross-validated classification correctly assigned 64.1% of the specimens, while classification success varies to a relatively large degree 20–96% between the groups (Table S3).

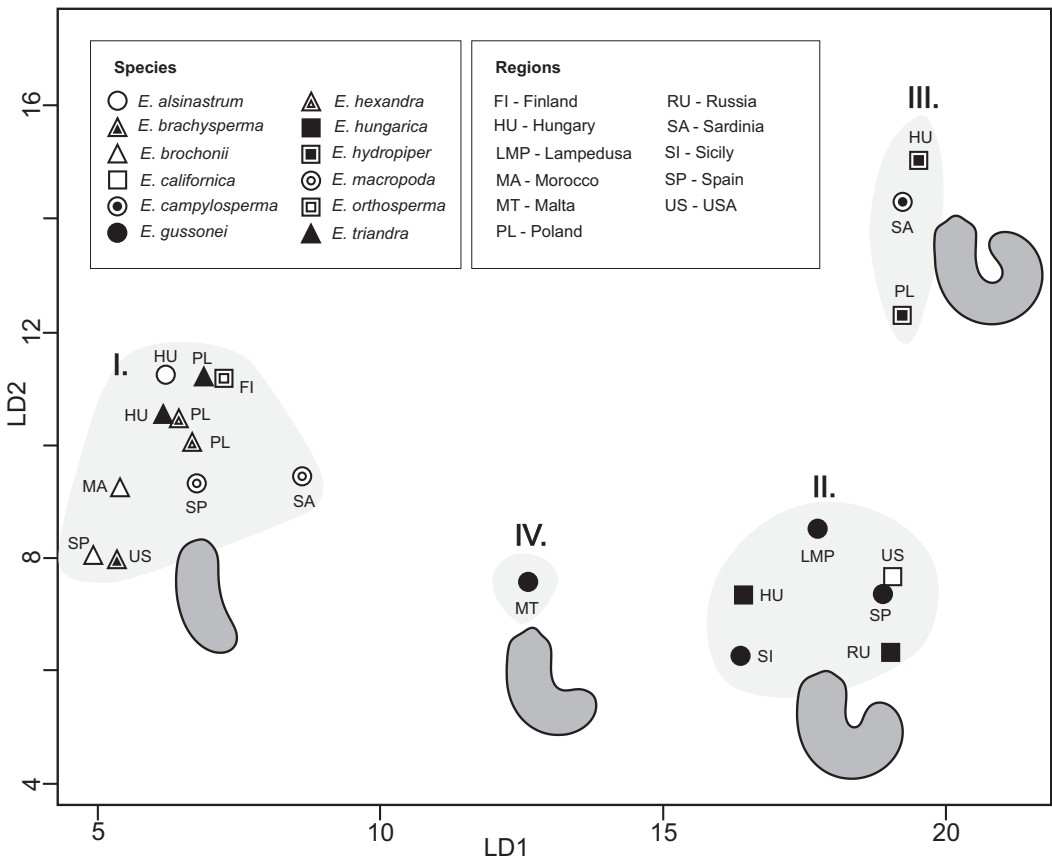

**Figure 5** **CVA scatter plot of seed outlines.** Average outlines are presented.

## DISCUSSION

In this paper we provide a detailed molecular phylogenetic study of the genus *Elatine* by sequencing and analysing the nuclear rITS region and four plastid regions in multiple accessions of 13 species. Through analysis of a combined nuclear and plastid data, the matrix has provided the most resolved and reliable phylogenetic hypothesis for the studied taxa, and is able to interpret the phylogenetic relationships hypothesized by this tree (Fig. 4).

### Molecular phylogenetic relationships

Although morphologically highly different (Fig. 1), *E. alsinastrum* was placed equivocally as sister to the rest of the genus. In contrast, *E. brochonii*, a trimerous flowered, haplostemon species of the Western Mediterranean was placed at the base of the tree. We have to note here that the analysis of *Cai et al. (2016)* placed this species unequivocally as sister to the rest of the genus, but that analysis lacked *E. brochonii*. The rest of the species analysed here are split into two main lineages, most probably representing the two main clades of the genus *Elatine*. One corresponds to the traditional section *Crypta*, and consists of trimerous flowered, haplostemon species with slightly or non-curved seeds (*E. ambigua*, *E. brachysperma*, *E. triandra*). The North American species thus apparently represent a separate lineage, and—together with the high species diversity of trimerous, haplostemon

species found on this continent—hint at the North American diversification and origin of the *Crypta* clade.

Within the diplostemon clade, *E. hexandra* branches first as sister to the rest; although this result is statistically highly supported, a stabilised hybrid origin of this species can be postulated based on its phylogenetic position. This placement of lineages could be indicative of hybrid origin, as discussed in detail by *Funk (1985)*, and is further corroborated by the presence of phylogenetically very distantly related ribotypes (Fig. 3) within the same individual, derived from the *Crypta* and the *E. brochonii* lineage (Fig. 2A). Furthermore, *Uotila (2009a)* reported the presence of morphological instability in this species, as three and exceptionally four carpels can be observed. Highest ploidy level (dodecaploidy) can also be found in this species (*Jankun, 1989*),perhaps the result of allopolyploidisation (*Wendel, 2000*), and could have prevented the completion of concerted evolution of nrITS arrays located on non-homologous chromosomes (*Wendel, Schnabel & Seelanan, 1995*; *Álvarez & Wendel, 2003*). Thus, on the basis of these arguments, we conclude an allopolyploid hybrid origin for the species *E. hexandra*. Although its exact origin is equivocal, *E. triandra* as one parent can be stated with high certainty, while high ploidy level in this species might be the sign of the involvement of more than one additional taxa. Additional ribotypes hint at an *E. brochonii*-like ancestral species (trimerous flowers, slightly curved seeds, haplostemon adroecium), while the plastid affinity to tetramerous members of the genus clearly indicates genome donors with tetramerous flowers.

The other main lineage within this section of the tree is represented by the tetramerous flowered, diplostemon species of section *Elatinella*. The species in this lineage can further be divded into two main sub-lineages: one consist of species with Temperate distribution (Fig. 4: clade D), while the other one includes species of Mediterranean distribution (Fig. 4: clade E). Out of these, all but *E. californica* (North American), *E. hungarica*, and *E. hydropiper* (both Palearctic) are confined geographically to the European continent, where the diversification of section *Elatinella* might have taken place. Nonetheless, the placement of *E. californica* close to the root of this clade can imply a Nearctic origin for the section; this group might have originated in North America and then later diversified in the Old World. However, this is in contrarst to the result of *Cai et al. (2016)*, who postulated a Eurasian origin of the genus. Within clade D, the northern Eurasian *E. orthosperma* is hypothesised to be sister to the species-pair *E. hungarica* and *E. hydropiper*, which show certain vicariance; the former is typical of the Eurasian steppe zone (*Lukács et al., 2013*), while the latter can be considered to be an Atlantic-boreal species inhabiting more northern latitudes in Eurasia (*Popiela et al., 2012*).

The other main sublineage (clade E) is represented by species inhabiting the Mediterranean Basin, including *E. campylosperma*, *E. gussonei*, and *E. macropoda*. The species branching the earliest is *E. campylosperma*, a taxonomically neglected species (see below). This is followed by *E. gussonei*, what is further split into two lineages, the western Mediterranean samples form a separate clade, while plants from Malta and Lampedusa are sister to a monophyletic *E. macropoda*. However, there are some notable incongruencies between the trees obtained from contrastingly inherited markers (Fig. 2): (i) including the Spanish *E. campylosperma* that has identical nrITS to *E. macropoda*, but shares

plastid haplotypes with the other *E. campylosperma* from Italy; (ii) including the Turkish *E. macropoda* which is included in the *E. gussonei* clade on the nrITS tree, nested within the *E. macropoda* clade on the plastid tree. These incongruent placements on trees of differently inherited markers are commonly explained by recent hybridisation (*Rieseberg, Whitton & Linder, 1996*; *Wendel & Doyle, 1998*). Indeed, these Mediterranean species of *Elatine* have much bigger and conspicuous flowers compared to their Temperate siblings (Fig. 1), which probably promotes cross-pollination by insects, opening the way to hybridisation of these species that often occur in sympatry. Indeed, the merging of *E. macropoda* and *E. gussonei* ribotypes in the Maltese *E. gussonei* accession directly demonstrates this capability of setting seed by cross-pollination in this predominantly selfing genus.

## Seed morphometrics in *Elatine*

On the one hand, our results clearly indicate that the outline of seeds alone is not suitable for species delimitation while, on the other, four morpho-groups can be recognized in the in the CVA plots among *Elatine* (Fig. 5), including straight, highly curved, curved, and an intermediate seed shape which are clearly identifiable. Based on the phylogenetic relationships among species, it seems that these main seed shapes do not form monophyletic units; the straight seed shape appears in both the earlier and the most recently divergent species, suggesting that they could have evolved multiple times during the evolution of the genus. Therefore, seed shape alone can only be used to define species within a given evolutionary lineage.

## Species delimitation in *Elatine*

The inclusion of multiple accessions of the same *a priori* species enabled us to test the species delimitations and specific characters used in the taxonomy of this genus. As demonstrated on the plastid tree (Fig. 2B) and on our combined tree (Fig. 4), in most cases multiple accessions of the same species were grouped into the same tip clade. This is one clear indication of the genetic cohesiveness of this species as interpreted in the current taxonomy of the genus (*Cook, 1968*; *Uotila, 2009b*), plus these tip clades are placed on rather long branches in phylograms (Figs. 2 and 4A) indicating substantial genetic differentiation. One notable exception, however, is the *E. hungarica*—*E. hydropiper* sibling species, where there is a significant difference in seed characteristics (Fig. 5). Similarly, no substantial genetic differences exist between our *E. ambigua* and *E. triandra* samples, corroborated by seed characteristics. Thus, given the certain identification of our *E. ambigua* sample—the presence of significant pedicels, the diagnostic character of this species (*Cook, 1968*)—we are confident these samples are taxonomically equivalent, thus questioning the presence of the Asian *E. ambigua* in Europe. In fact, our field experience also suggest this as we repeatedly found *E. triandra* specimens—usually in full sunshine—with long pedicels in Hungary. This probably also explains why there are scattered and ephemeral observations of this species in Europe (*Moesz, 1908*; *Cook, 1968*).

Probably the most conspicuous discrepancy in species delimitation concerns *E. gussonei*, a neglected species of the Mediterranean Basin (*Mifsud, 2006*; *Kalinka et al., 2014*). Our samples of this species fall into two distinct clade (Fig. 4), one which includes the sample

from the nomenclatural type locality of Lampedusa (*Molnár, Popiela & Lukács, 2013b*) is sister to *E. macropoda*, while the other, including mostly Western Mediterranean samples, forms a separate monophyletic lineage. This substantial genetic difference has to be further studied, but most probably merits taxonomic recognition of at least the subspecies level.

Finally, the presence of an almost forgotten species of European *Elatine*, *E. campylosperma*, should be discussed. This plant was described by *Seubert (1842)* from Sardinia, but was later neglected by most workers who synonymized it under *E. macropoda* (*Cook, 1968*; *Uotila, 2009b*; *Popiela & Łysko, 2010*). Our data show that plants with highly curved seeds and conspicuous flowers from the Mediterranean Basin can be distinguished as separate molecular and morphological entities, and our ongoing taxonomical investigation suggests that these should be treated as *E. campylosperma*.

## Taxonomic implications

On the basis of the well-resolved molecular phylogenetic tree presented in this paper (Fig. 4), we test the currently used systematic treatment of the genus *Elatine* proposed originally by *Seubert (1845)*. We corroborate the earlier observation that section *Crypta* is monophyletic, although our sampling was not focused on this group. In contrast, the other section *Elatinella* was found to be polyphyletic, including the species *E. brochonii*. Disregarding this species, the section is monophyletic, further demonstrating the utility of floral morphological characters used in the systematics of *Elatine*.

Nevertheless, more details are unravelled in this study regarding the intra-sectional genetic lineages and chromosome number characteristics of *Elatine* section *Elatinella* (summarised on Fig. 4). Based on these results, we devise the following new section corresponding to clade A, and two new subsections corresponding to clade D and clade E (on Fig. 4):

### *Bracteata Sramkó, A. Molnár & Popiela, sect. nov.*

Type: *E. brochonii* Clav.

Morphology—Stems 1–10 cm long, erect or prostrate, leaves elliptical, oval or oblong; leafy bracts; two-five trimerous, diplostemonous, sessile flowers in cyme; the supreme flower single, six stamens; seeds straight.

Diagnostic characters—leafy bracts; the supreme flower single; short, straight seeds.

Etymology—The section was named on the basis of the significant and characteristic bracts of the only species.

Distribution—Western Mediterranean (Morocco, Algeria, Spain, Portugal, France, Corsica).

Accepted species—*Elatine brochonii.*

### *Hydropiperia Sramkó, A. Molnár & Popiela, subsect. nov.*

Type: *E. hydropiper* L.

Morphology—Shoots branched, rooting at nodes and creeping, 1-10 cm long; leaves oblong, ovale or spatulate; one-two flowers per node, axillary, diplostemonous, tetramerous, sessile or short pedicelled (elongating in fruit), eight stamens; seeds horseshoe, crescent-shaped or long and straight (*E. orthosperma*).

Diagnostic characters—Procumbent and node-rooting plants; short or very short pedicels, mostly moderate zone, blooms summer/autumn (VI-X).

Etymology—This subsection was named after its most widespread species (*E. hydropiper*).

Distribution—Circumboreal (Eurasia, North-America).

Accepted species—*Elatine californica, E. hungarica, E. hydropiper,* and *E. orthosperma.*

### Macropodae Sramkó, A. Molnár & Popiela, subsect. nov.

Type: *E. macropoda* Guss.

Morphology—Plants about 10 cm long with long internodes, most frequently upright stem, sometimes rooting at nodes and creeping; leaves obovate or oblong obovate; diplostemonous, tetramerous flowers on long pedicels, usually one per node; eight stamens; seeds slightly curved, comma-shaped or horseshoe.

Diagnostic characters—Long pedicel, usually erect; Mediterranean zone; blooms winter/spring (I–IV).

Etymology—This subsection was named after *E. macropoda*.

Distribution—Mediterranean: mainly on coastal zones and archipelago of the Mediterranean Sea.

Accepted species—*Elatine campylosperma, E. gussonei, E. macropoda.*

## ACKNOWLEDGEMENTS

We gratefully acknowledge the field assistance of Endre Bajka, Bartosz Kurnicki, Gergely Gulyás, Gusztáv Jakab, Andrzej Łysko, Viktor Löki, Attila Mesterházy, Edvárd Mizsei, Ágnes Mosolygó-Lukács, Arkadiusz Nowak, Sylwia Nowak, Norbert Pfeiffer, Lajos Somlyay, Antal Széll, László Tóth, and Róbert Vidéki, as well as the laboratory assistance of Kaan Hürkan, Ágnes Mosolygó-Lukács. Thank the staff of the Centre for Molecular Biology, University of Szczecin–Bożena Białecka and Magdalena Bihun—for taking the scanning electron micrographs. We appreciate the work of our reviewers, and are very grateful to Gareth Dyke for his improvements to the English of our work.

### Funding

This research was supported by the European Union and the State of Hungary, co-financed by the European Social Fund in the framework of TÁMOP-4.2.4.A/2-11/1-2012-0001 'National Excellence Program.' Instrumental and infrastructural support was received from OTKA K108992 Grant (AMV), OTKA PD109686 (GS) and the National Science Center (Poland) N N303 470638 Grant (AP). This work was supported by the János Bolyai Scholarship of the Hungarian Academy of Sciences. The funders had no role in study design, data collection and analysis, decision to publish, or preparation of the manuscript.

## Grant Disclosures

The following grant information was disclosed by the authors:
European Union and the State of Hungary: TÁMOP-4.2.4.A/2-11/1-2012-0001.
OTKA: K108992, PD109686.
National Science Center (Poland): N N303 470638.
Hungarian Academy of Sciences.

## Competing Interests

The authors declare there are no competing interests.

## Author Contributions

- Gábor Sramkó conceived and designed the experiments, performed the experiments, analyzed the data, contributed reagents/materials/analysis tools, wrote the paper, prepared figures and/or tables, reviewed drafts of the paper.
- Attila Molnár V. conceived and designed the experiments, contributed reagents/materials/analysis tools, wrote the paper, prepared figures and/or tables, reviewed drafts of the paper.
- János Pál Tóth analyzed the data, contributed reagents/materials/analysis tools, wrote the paper, prepared figures and/or tables, reviewed drafts of the paper.
- Levente Laczkó performed the experiments, reviewed drafts of the paper.
- Anna Kalinka, Orsolya Horváth and Lidia Skuza performed the experiments, contributed reagents/materials/analysis tools, reviewed drafts of the paper.
- Balázs András Lukács contributed reagents/materials/analysis tools, reviewed drafts of the paper.
- Agnieszka Popiela conceived and designed the experiments, contributed reagents/materials/analysis tools, reviewed drafts of the paper.

## Field Study Permissions

The following information was supplied relating to field study approvals (i.e., approving body and any reference numbers):

*Elatine hungarica*, *E. hydropiper* and *E. triandra* are protected species and were sampled in Hungary with the permission of the Hortobágy National Park Directorate (Permission id.: 45-2/2000, 250-2/2001).

## DNA Deposition

The following information was supplied regarding the deposition of DNA sequences:
GenBank accession numbers are provided in Table 1.

## Data Availability

The raw data has been supplied as a Supplemental Dataset.

## Supplemental Information

Supplemental information for this article can be found online at http://dx.doi.org/10.7717/peerj.2800#supplemental-information.

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
