# Peer review of "Molecular phylogenetics, seed morphometrics, chromosome number evolution and systematics of European Elatine L. (Elatinaceae) species"

_PeerJ, doi:10.7717/peerj.2800_

## Round 0.1 · original submission · Minor Revisions

Both reviewers agree that the study merits publication in PeerJ, but both have concerns that should be addressed before the article can be accepted. Please review the writing to ensure clarity, correct grammar, and remove typos. I also agree with Reviewer 1's concerns regarding polyploidy, and this reviewer makes a number of other salient points including discussion of the tree topology and polymorphism in markers used.

Reviewer 1 ·

Basic reporting

The introduction in thorough and clearly outlines the background information on the systematics of the genus. The English is understandable but a bit awkward in a few places. A careful review by a native speaker would help. I have listed some typos in my general comments below, but certainly not all.

Experimental design

No comments. The experimental design is adequate.

Validity of the findings

No comments.

Additional comments

The authors present a molecular phylogenetic study of the genus Elatine, based on DNA sequences of ITS and on 4 plastid loci, with focus on the European species, and they compared the results to a morphological study of seed shape characters, which have traditionally been used in species delimitation in the genus. They base several taxonomic conclusions on the phylogenetic results, but conclude that the use of seed shape characters, while useful to define broad groups, is less useful for species delimitation.

As noted above, the methods are sound in general. The English is understandable but a bit awkward in a few places. A careful review by a native speaker would help. I have listed some typos below, but certainly not all.

I list several suggestions below that I believe would improve the manuscript. Most importantly, I think that polyploidy should be discussed more explicitly, not only with respect to the likely allopolyploidy origin of E. hexandra, but in the genus more broadly. There is a wide range of ploidy levels among the sampled taxa, so this issue could be discussed in a separate section. The possible effects of polyploidy on the differences among the ITS and plastid trees could be mentioned, as well as the possible evidence for the genome donors of other species besides just E. hexandra could be discussed.

The following are some suggestions and concerns, which would improve the manuscript if addressed. These comments are listed by line number, or later by the Table or Figure number.

Lines 70-73: The manuscript states that there are 25 species in the genus overall, and in this study 13 species were sampled, with all of the European species included. This would imply that 12 species (nearly half) were not included. I assume most of these 12 unsampled species were in section Crypta, although that isn’t stated explicitly at lines 70-73. It would be helpful to list the total number of species in that section when it is described, so that readers don’t need to search the manuscript to infer that information.

94-95: “none of these were focusing on the internal phylogenetic relationship of it.”
expand to: “none of these were focusing on the internal phylogenetic relationships within the genus.”

106: “provide a modern systematic [treatment] based on our results”

Lines 112-113, also include numbers here - # European, # North American, etc. (even if in table) four species of section Crypta out of how many? (see above)

Line 137, “ground” is already past tense, so “grounded” is not appropriate here.

139: typo: "than" change to "then"

135-144: This is a fairly standard CTAB extraction protocol, so why not just cite previous work (e.g., Doyle & Doyle 1987 or 1990)?

175: typo: "than" change to "then"

182-183: reword either to
“In case of nrITS sequences a [careful screening] for additive polymorphic sites (i.e. overlapping peaks at certain sites) was performed…”
or
“[The] nrITS sequences [were] carefully screened for additive polymorphic sites (i.e.
183 overlapping peaks at certain sites),”

186-188 (how many differences between the two parental sequences? No indels, I assume, or it would have been unreadable.

212-213: How many samples had to be excluded due to paralogy among ITS sequences?

218: Preferred tree? Preferred on what basis? (see my comment below at the Figure)

272-274: I am aware of other taxa for which trnH-psbA shows length variation, but never within a single sample. Given that this is a plastid locus and that multiple haplotypes were not observed for other plastid loci, what is your interpretation of the possible cause of this observation? Might it be cross-contamination of samples?

279: How does this tree differ from the other equally parsimonious tree?

282: Wasn’t Bergia texana defined as the root a priori? The results of parsimony analyses are generally displayed with the root as defined by the user.

283-284: “although placed sister to the rest of the genus, the sister relationship of Elatine alsinastrum is not supported” I think you mean that although results of the parsimony analysis support it as sister to the rest of the genus, that sister relationship does not have bootstrap support or support in the Bayesian analysis. (That is, unless it was not resolved in the alternative parsimony tree, in which case it wasn’t even supported by the strict consensus of the parsimony results.)

324-325: “E. hexandra is placed as sister to E. brochonii (although with weak support) on the nrITS tree” but that only describes one of the places it is placed on the ITS tree, so the statement is incomplete here. Later a more complete description is given of the placement of the sequences of that species in more than one place on the ITS tree.

367-370: Speaking of one species as equivocally sister to the rest of the genus but then saying another species is “closest to the root of the tree” doesn’t make sense.
Remember that parsimony analyses are correctly shown with an unresolved tricotomy at the base. This is because the placement of particular changes as being either on the branch to the outgroup or along the branch from the root to the ingroup cannot be resolved. This means that the outgroup will never be resolved as unequivocally resolved unless there are further outgroups included.

377-382: Discussion of E. hexandra as a “stabilized hybrid” seems overly complicated and dances around the idea of allopolyploidy instead of beginning with that idea. Given its high ploidy level, and placement of its ITS clones in different parts of the ITS tree, isn’t the simplest explanation allopolyploidy? Is there another kind of stabilized hybrid origin that would be more likely that allopolyploidy in this case? Why invoke anagenesis of an ancestor close to E. brochoni? Morphological differences between an allopolyploid and its progenitor species would not be unexpected. This species is described here as dodecaploid, so this could explain why its ITS sequences are placed with two potential progenitor species groups, whereas the plastic tree places the sequences from this species in a completely different part of the tree. There seems to be evidence of at least three genome donors of this high polyploid among the ITS and the plastid trees. The placement “between lineages” in the combined tree may be a result not only of hybrid origin but of combining characters derived from different progenitors of the hybrid (allopolyploid). I would suggest reorganizing this paragraph to begin with the high ploidy level of this species, followed by each kind of evidence that was found that supports the hypothesis of allopolyploidy and the various forms of evidence for the potential ancestors of the genomes of this polyploid species.

This species is only the highest of several high polyploids, so the phylogenetic results should be discussed further in the context of polyploidy. Both the evidence provided by the results for the origins of the polyploids, and also the potential effects of polyploidy on the results.

396: typo calde --> clade

398-400: Please clarify. Do you mean that the section might have originated in North America and then later diversified in Europe, or do you mean that there is equivocal evidence for an origin and diversification in either of these continents?

410-422: Recent hybridization in only one of several explanations for incongruous results with different, independently inherited loci. Hybridization (including chloroplast capture and other kinds of introgressive hybridization is one possibility, but so are incomplete lineage sorting and mistaking orthology and paralogy (which can be further complicated by homeologous variants, which is potentially an issue with the polyploids here). Be careful not to jump to one of several possible conclusions.

448: typo: filed --> field

496: typo: See -->Sea

Tables and Figures:

Table 1: Locality for Bergia should also include California as the state, rather than just the city and USA.

Table 2: I’m not sure that Table 2 is necessary, if the only point is to show the other loci that were assessed but not selected for the study. At the most, a list of other loci that were assessed but not selected for the full study could be included.

Fig. 2. Are the branches that as shown as dashed lines because they “lack statistical support” also the branches that collapse in the consensus tree of the maximally parsimonious trees? These are described as “preferred” trees among the maximally parsimonious trees, but I didn’t see a description of the criteria that made them “preferred” or how the other MP trees differed from these ones. Am I missing something? Was this based on the bootstrapping values, or on the Bayesian analysis?

Fig. 4. Given that clones of E. hexandra PL2 were placed in two places on the ITS tree, how was the one selected to represent this plant (population?) in the combined analysis? Given that one of the clones was selected instead of the other, this should be justified.

·

Basic reporting

The text was fairly well written but there are some obvious typographic errors that need to be addressed, as well as some grammar issues. These are not big problems and I assume these will be fixed during the editing process. It appears that relevant prior literature was included and cited. All figures and tables are appropriate and sufficient to convey the presented data.

Experimental design

The manuscript follows protocols that are standard in the field and the authors sufficiently defined the research question and experimental design. This was the first publication of a molecular framework for Elatine and morphological analysis was also included, which bolstered the results. Based upon the methods presented, it should be possible to easily duplicate the work.

Validity of the findings

The data was robust. These analyses are fairly straightforward and are sound. As long as the sequences are included in GenBank (which they state will happen), then the data is provided in a publicly accessible, and acceptable, database. The results of the morphometric analysis are provided as supplemental material.
When there are discrepancies between molecular phylogenies, the authors did a good job speculating on the reasoning for the discrepancies. This is common in molecular phylogenetics and the authors did a fine job explaining where discrepancies occurred and why they might have occurred.

Additional comments

There are minor editorial issues to be resolved including a few grammar and spelling errors. These should be easy to fix. Usually, the errors were spelling issues like 'brat' instead of 'bract', and errors relating to the use of 'is' and 'are'. However, please pay special attention to the spelling of the scientific names as they do not always match within the document (example: borchoni and brochonii). I assume that these are the same taxa but an error like this should not happen in a submitted manuscript. I did not check the spelling for every name to be sure that it matched throughout the manuscript so take care to check these again.

---

## Round 0.2 · Minor Revisions

Thank you for your attention to reviewer concerns. Three final minor revisions:

First, it is not clear whether you addressed the reviewer's concern that "Speaking of one species as equivocally sister to the rest of the genus but then saying another species is “closest to the root of the tree” doesn’t make sense...". Some of the original text appears changed, but is this comment not also relevant for sentences such as "E. californica close to the root of this clade implies a Nearctic origin for the section;"?

Second, I agree with the reviewer's concern and confusion about "preferred" tree and think the term "arbitrary" would better reflect that this was a choice not based on support from data/statistics.

Finally, I would go further than the reviewer -- I think tables 2-4 are lengthy and not particular useful to the reader in the main text. While this is an online only journal, efforts to make the main text more readable are nonetheless a good idea.

---

## Round 0.3 · accepted · Accept

Thank you for addressing previous concerns; the addition of Cai et al also seems very appropriate and well done. Please do take this chance to revise grammar/typing etc. throughout (e.g. line 296, 236 etc).